# A Novel Bone Substitute Based on Recombinant Type I Collagen for Reconstruction of Alveolar Cleft

**DOI:** 10.3390/ma14092306

**Published:** 2021-04-29

**Authors:** Masaaki Ito, Taku Toriumi, Takahiro Hiratsuka, Hideto Imura, Yasunori Akiyama, Ichinnorov Chimedtseren, Yoshinori Arai, Kazuhiro Yamaguchi, Akihiko Azuma, Ken-ichiro Hata, Nagato Natsume, Masaki Honda

**Affiliations:** 1Division of Research and Treatment for Oral and Maxillofacial Congenital Anomalies, School of Dentistry, Aichi Gakuin University, Aichi 464-8651, Japan; masa1119mile@gmail.com (M.I.); h-imura@dpc.agu.ac.jp (H.I.); y.akiyama.agud0626@gmail.com (Y.A.); Ichko.dentist@gmail.com (I.C.); natsume@dpc.agu.ac.jp (N.N.); 2Department of Oral Anatomy, School of Dentistry, Aichi Gakuin University, Aichi 464-8650, Japan; toriumi@dpc.agu.ac.jp; 3Bio Science & Engineering Laboratory, Research & Development Management Headquarters FUJIFILM Corporation, Kanagawa 258-8577, Japan; takahiro.hiratsuka@fujifilm.com (T.H.); kazuhiro.yamaguchi@fujifilm.com (K.Y.); akihiko.azuma@fujifilm.com (A.A.); ken-ichiro_hata@jpte.co.jp (K.-i.H.); 4Department of Oral and Maxillofacial Radiology, Nihon University School of Dentistry, Tokyo 101-8310, Japan; mhg01033@nifty.com

**Keywords:** alveolar cleft, bone reconstruction, collagen scaffold, cross-link density, histological analysis, human collagen type I, micro-CT, rat palatine fissure, recombinant peptide RCP

## Abstract

This study aimed to examine the optimal cross-link density of recombinant peptide (RCP) particles, based on human collagen type I, for bone reconstruction in human alveolar cleft. Low- (group 1), medium- (group 2), and high- (group 3) cross-linked RCP particles were prepared by altering the duration of the heat-dependent dehydration reaction. Rat palatine fissures (*n* = 45), analogous to human congenital bone defects, were examined to evaluate the potential of bone formation by the three different RCP particles. Microcomputed tomography images were obtained to measure bone volume and bone mineral density at 4, 8, 12, and 16 weeks post grafting. Specimens were obtained for histological analysis at 16 weeks after grafting. Additionally, alkaline phosphatase and tartrate acid phosphatase staining were performed to visualize the presence of osteoblasts and osteoclasts. At 16 weeks, bone volume, bone mineral density, and new bone area measurements in group 2 were significantly higher than in any other group. In addition, the number of osteoblasts and osteoclasts on the new bone surface in group 2 was significantly higher than in any other group. Our results demonstrated that medium cross-linking was more suitable for bone formation—and could be useful in human alveolar cleft repairs as well.

## 1. Introduction

Bone reconstruction of the alveolar cleft is one of the most difficult and challenging alveolar bone defect repairs. Undoubtedly, autologous bone graft is widely regarded as the gold standard for this purpose [1,2,3,4]. However, it has several drawbacks, including pain at the donor site, development of deformity, possible injury to adjacent anatomical structures, and loss of grafted bone [5,6]. Development of a better strategy is recommended. Bone is a dynamic mineralized tissue, composed of organic extracellular matrix (ECM) and inorganic minerals, supporting the body framework and ensuring mineral homeostasis in body fluids. The ECM of bone consists of collagen and a lesser amount of noncollagenous proteins, lipids, and water [7]. Type I collagen is the principal component of the organic matrix of bone, as well as of other connective tissues [8,9]. Therefore, type I collagen-based bone substitute has received significant attention in the field of bone tissue engineering [10]. Natural type-I collagen has been reported to show enhanced cellular attachment, and induce osteogenic differentiation, owing to its abundant Arg-Gly-Asp (RGD) residues [11,12,13]. Recently, several studies have focused on the use of cell adhesion motifs containing RGD on the bone substitute surface, as incorporation of RGD-containing peptides on the bone substitute could reinforce cell adhesion and osteogenic differentiation [14]. RGD residues constitute an important factor for the generation of bone tissue in collagen-based bone substitute. Numerous studies have shown that incorporation of RGD-containing peptides on biomaterial surfaces increases cell adhesion and osteogenic differentiation [15,16], because RGD-peptides facilitate the interaction of cellular integrin receptors with bone matrix proteins [17,18,19]. Another study showed that coating HA surfaces with an RGD-containing peptide increased attachment and differentiation of osteoblasts [20,21,22]. Altogether, existing evidence suggests that a bone substitute containing RGD peptide is useful for the treatment of large bone defects.

Some major drawbacks of the current collagen-based bone substitute include limited mechanical strength and rapid biodegradation. The biochemical and biophysical properties of collagen-based bone substitute are known to greatly affect cell behavior, including survival, proliferation, and differentiation, due to diverse mechanical strength [23,24]. For example, the cellular response to monomeric or denatured collagen differs from the response to naturally generated collagen fibrils [25,26]. Monomeric collagen stimulates the proliferation of arterial smooth muscle cells, in comparison with the polymerized type I collagen [27]. Additionally, matrix elasticity affects the differentiation of osteolineage cells and mesenchymal stem cells (MSCs) [28]. Altogether, collagen-based bone substitute requires an optimal cross-link density for regulation of its properties, since the extent of collagen cross-linking affects cellular activity, fibrillogenesis, matrix stability, and elasticity of bones [29,30].

Fujifilm Corp. (Tokyo, Japan) developed a novel bioabsorbable recombinant protein (RCP), based on the alpha-1 sequence (α I chain) of human collagen type I (having 12 RGD motifs in a single molecule and a repeat of the RGD peptide) for medical applications [31,32,33,34,35,36,37]. RCP, produced by the yeast *Pichia pastoris*, differs from conventional animal collagen in that there is no risk of infection (e.g., bovine spongiform encephalopathy) associated with it [31,32]. Additionally, RCP has several features as a bone substitute. First, the RCP is biodegradable and bioabsorbable—it does not remain in the body. Second, RCP has good biocompatibility as cellular scaffolding [33,34]. To date, a series of studies have been performed to elucidate the characteristics of RCP. Pawelec et al. showed that human MSCs could proliferate and differentiate on RCP [33]. BMSC-RCP combination sponge constructs were found to promote functional recovery post-implantation onto the ipsilateral intact neocortex for ischemic stroke [34]. RCP also shows great potential for angiogenesis. When constructs of RCP sponge and MSCs were subcutaneously implanted into mice, MSCs were found to accelerate angiogenesis in the graft [32]. Additionally, Mashiko et al. showed that RCP sponges could enhance the function of human adipose-derived stem cells in wound healing [35]. Finally, this material is capable of controlling cross-link density and changing the shape of RCP-based bone substitutes [36,37]. Despite this, there have been no reports to date regarding the optimal cross-link density of RCP-based bone substitute for bone reconstruction in the human alveolar cleft. The purpose of this study was to develop a novel bone substitute with RCP particles, with optimal cross-link density. Materials were evaluated for bone formation potential by implanting them into rat palatine fissures.

## 2. Materials and Methods

### 2.1. Animals and Housing

All experiments were performed using 11-week-old healthy male Sprague–Dawley rats with a body weight of 350–400 g (Chubu Kagaku Shizai, Nagoya, Japan). All rats were housed at an animal experimentation laboratory under standardized temperature and humidity, with a 12-h day/night cycle, at the Animal Research Center of Aichi Gakuin University. The study protocol was approved by the Animal Care and Use Committee of the School of Dentistry, Aichi Gakuin University (approval no. AGUD412). Animal care and experimental procedures were conducted in accordance with the Regulation on Animal Experimentation at School of Dentistry, Aichi Gakuin University.

### 2.2. Preparation of RCP Particles with Different Densities

The recombinant peptide (RCP) from human type I collagen α chain was prepared as described previously. Porous sponge blocks (formed by freeze drying of RCP solution) were cross-linked via a heat-dependent dehydration condensation reaction and crushed into particles. Three types of RCP particles were prepared with three different cross-link densities by altering the duration of the heat-dependent dehydration condensation reaction. The durations were: 3.5 h for low cross-linking, 4.75 h for medium cross-linking, and 7 h for high cross-linking. The average diameter of the mRCP particles was approximately 1000 μm, and ranged from 1058 to 1133 μm.

### 2.3. Water Absorption Rate and Acid Decomposition Residual Rate

Next, the water absorption rates and acid decomposition residual rates of all three types of RCP particles were determined. Fifteen milligrams of the test substance and 1.7 mL of 1 mol/L hydrochloric acid solution were put in each tube (whose tare weight had been measured in advance). These were then allowed to stand at 37.0 °C for 3 h before being transferred to ice and centrifuged. After centrifugation, the supernatant was removed, leaving the precipitated test substance behind. The latter was washed with water and centrifuged again. The precipitated test substance was freeze-dried, and the container with the test substance was weighed after drying the content (total weight after reaction). Weight of the test substance after the reaction was divided by that before the reaction to obtain the residual rate of acid decomposition (%). Residual rate of acid decomposition (%) = (total weight after reaction − tare weight)/weight of test substance before reaction × 100 (Table 1).

Water absorption rate was calculated by the following method: ten milligrams of the test substance were collected in each filter cup (with a volume of 500 μL, equipped with a filter of pore size 0.22 μm at the bottom, hereafter referred to as a container), whose tare weight had been measured in advance. Sufficient water was added and mixed until the test substance was saturated with absorbed water. The excess water was removed by centrifugation, and the container with the test substance (with absorbed water) was weighed. A blank test was performed separately; the value obtained by subtracting the tare weight from the total weight after water absorption was the residual water taken when the test substance was not included. The average value of the volume of residual water was taken as the residual water volume of the blank test, and the water absorption rate was calculated accordingly. The weight of the test substance after water absorption was divided by that before water absorption to obtain the water absorption rate (%). Water absorption rate (%) = (total weight after water absorption − tare weight − remaining water in blank test)/weight of test substance before water absorption × 100 (Table 1).

### 2.4. Pore Size of RCP Particles

All three types of RCP particles were sprayed with an approximately 30 nm layer of carbon using a vacuum evaporator (JEE-420T; JEOL, Tokyo, Japan). The pore size within the RCP particles was evaluated with a field-emission electron probe microanalyzer (EPMA, JXA-8530FA, JEOL) (Figure 1A–C). Six pores were randomly selected. Based on their diameters, the average pore size of the RCP particles was calculated (Table 1).

### 2.5. Preparation of Graft Bed Created in Left Palatine Fissure of Rat Maxilla

A total of 45 rats with palatine fissures on the left side were randomly subdivided into three groups, based on the implants administered: group 1 received low cross-linked RCP particles, group 2 received medium cross-linked RCP particles, and group 3 received high cross-linked RCP particles.

The process of creating the graft bed in the left palatine fissure was described in detail in our previous study [38]. Briefly, the surgery was performed under sterile conditions following general anesthesia. A 10 mm longitudinal incision was made along the reflection between the body of the left incisive bone and the palatine process of the maxilla, and the periosteal flap between palatine process and nasal septum was elevated. The RCP particles were immediately placed in the graft bed created in the palatine fissure. In the control group, a blank defect in the right palatine fissure was left untreated.

### 2.6. Micro Computed Tomography (Micro-CT) Imaging and Analysis of Hard Tissue Formation

The three rat groups (*n* = 15 each) were evaluated by micro-CT analysis. In vivo X-ray micro-CT (Cosmo Scan GX; Rigaku Corporation, Tokyo, Japan) was used for imaging, as previously described [39,40]. The exposure parameters were: 18 s, 90 kV, and 100 μA. The isotropic voxel size was 45 μm. Images of newly formed hard tissue were obtained from each rat at 4, 8, 12, and 16 weeks after surgery.

Bone volume and BMD were measured in the regions of interest (ROIs) from voxel images using the bone volume-measuring software 3 by 4 viewer 2011 (Kitasenjyu Radist Dental Clinic i-VIEW Image Center, Tokyo, Japan). The ROI size was 1.8 × 2.7 × 0.9 mm^3^, which covered the area used for RCP particle grafting. In addition, two reference points were selected as the centers of the ROI in the horizontal plane and the frontal plane, individually, to obtain the same ROI position after RCP particles were grafted (Figure 2). To obtain a proper horizontal plane image, the reference point was set 6–7 mm from the center of the incisor toward the molar. To obtain a proper frontal plane image, the reference point was set at a depth of 1.5–2 mm from the apex of the nasal septum (Figure 2). Bone volume and BMD in the ROI were measured before grafting and 4, 8, 12, and 16 weeks after grafting. The increases in bone volume and BMD in individual rats were calculated by subtracting the value measured before grafting from the values measured at 4, 8, 12 and 16 weeks after grafting.

### 2.7. Histological Analysis

Histological analysis was carried out at 16 weeks after grafting in the rats from the three groups. The animals were sacrificed in a carbon dioxide bath, and the harvested specimens were fixed in 4% paraformaldehyde for 24 h, decalcified in 10% ethylenediaminetetraacetic acid disodium salt (Muto Pure Chemicals, Tokyo, Japan) for 8 weeks, dehydrated through a graded series of ethanol solutions, and then embedded in paraffin. Specimens were prepared as horizontal plane sections (5 μm thick) with a microtome (Leica RM2165; Leica Microsystems, Nussloch, Germany), and the paraffin sections were stained with hematoxylin and eosin (H&E). The newly formed bone area was measured using ImageJ (National Institutes of Health, Bethesda, MD, USA).

Alkaline phosphatase (ALP) and tartrate acid phosphatase (TRAP; Sept. Sapie Cp., Ltd., Tokyo, Japan) staining was performed to analyze the osteoblasts and osteoclasts, respectively, on the new bone surface formed in the palatine fissure at 16 weeks after RCP particle grafting. The average number of osteoblasts and osteoclasts were counted from five histologically stained specimens each.

### 2.8. Statistical Analysis

Data are expressed as the mean and standard deviation for each group. Statistical analysis was performed using Excel Statistical File software (ystat2008.xls; Igakutosho-Shuppan Ltd., Tokyo, Japan). One-way analysis of variance with a Tukey–Kramer post-hoc test was used for intergroup comparisons. Differences showing *p* values < 0.05 were considered to be statistically significant.

## 3. Results

### 3.1. Clinical Results

The surgical procedures were well-tolerated by all rats. No wound dehiscence, severe inflammation, or swelling was observed in any of the samples throughout the experimental period. The animals showed no weight reduction during the healing time. The grafted RCP particles did not leak out during the experiments in any group (data not shown).

### 3.2. Physical Properties of the Three Types of RCP Particles

EPMA images of the three types of RCP particles and their physical properties are shown and summarized in Figure 1. EPMA images confirmed the wide range of pore sizes in the three types of RCP particles and revealed that many of the smaller pores were not interconnected (Figure 1A–C). The pore sizes in low cross-linked RCP particles ranged from 68 to 96 μm, with an average of 87.2 μm. The pore sizes in medium cross-linked RCP particles ranged from 74 to 99 μm, with an average of 86.4 μm. The pore sizes in high cross-linked RCP particles ranged from 66 to 103 μm, with an average of 86.8 μm (Table 1). There was no significant difference among the three types of RCP particles in terms of pore size. In addition, the total porosity of the three types of RCP particles was calculated; the mean porosity in all groups was approximately 80%.

Water absorption rate and acid decomposition residual rate are summarized in Table 1. With respect to these two parameters, differences were observed among the three types of RCP particles ((Table 1).

### 3.3. Micro-CT Analysis on Newly Formed Hard Tissue after RCP Particle Grafting

No opacity was observed on the X-ray images on the day after RCP particle grafting. Thereafter, opaque parts were observed in the palatine fissure at 4 weeks after grafting in all groups (Figure 3). Although the area of the opacity increased from 4 to 16 weeks, the overall size in group 1 was less than in groups 2 and 3. The bone volume and BMD measured using the opaque images are shown in Figure 4. Bone volume and BMD in group 1 were significantly lower than those in the other groups at 4, 8, 12 and 16 weeks. There were no significant differences in bone volume between groups 2 and 3 up to 12 weeks, but the bone volume in group 2 was significantly higher than that in group 3 at 16 weeks. In addition, there were significant differences in the BMD between groups 2 and 3 at 16 weeks, but not at 4, 8, or 12 weeks.

### 3.4. Histological Analysis of Newly Formed Bone

The H&E-stained histological images showed that bone tissue formation was observed in the areas corresponding to the opacities in the palatine fissures at 16 weeks after grafting in all groups (Figure 5A–C). A particularly obvious bone bridge was observed in the graft site in group 2 (Figure 5B). No remaining RCP particles were observed in the specimens in any group at 16 weeks (Figure 5A–C). The width of the palatine fissure cavity (graft side) was narrowed by the newly formed bone tissue in all groups, compared with that of the right palatine fissure (non-graft side). The cavity spaces were quite different between the three types of RCP grafts. The area of the newly formed bone (dashed line area) in group 2 was significantly higher than that in the other groups (Figure 5D). In group 2, higher magnification images showed lamellar bone and haversian canals (Figure 5F); however, newly formed bone appeared as woven bone in groups 1 and 3 (Figure 5E,G). In addition, many cells were present in the lacunae in all groups, which is characteristic of the presence of osteocytes in newly formed bone tissue.

A large number of cells lined the surface of the newly formed bone tissues, but the cell shapes were quite different among the three groups. The cells in group 2 showed a cuboidal/polygonal shape, while the cells in groups 1 and 3 showed a flattened shape (Figure 6A,D,G). The ALP staining intensity was stronger in group 2 than in groups 1 and 3 (Figure 6B,E,H). There were also some multinucleated giant cells observed on the surface of the newly formed bone in all groups. Cells stained with TRAP were located in Hawship’s lacuna (Figure 6C,F,I). The average number of osteoblasts and osteoclasts on the new bone surface in group 2 was significantly higher than that in the other groups (Figure 6J,K).

## 4. Discussion

This study evaluated the use of RCP particles as a prospective therapeutic approach for bone reconstruction to treat human alveolar cleft, a congenital defect. Why was the particle-like shape of RCP selected to repair the alveolar cleft? In our previous study, RCP blocks (Fujifilm Corp.) were grafted into an artificially created large bone defect at the inferior border of the rat mandible. Newly formed bone was clearly observed at the defect site at 4 weeks after grafting. The results demonstrated that RCP blocks are useful for repairing large bone defects [36]. However, the block shape is not compatible with the anatomical form of human alveolar cleft. It is easy to surgically carry particles to the depth of the alveolar cleft. Therefore, in this study, the shape of the RCP-based bone substitute was changed from block-type to a particle. To the best of the authors’ knowledge, no studies have evaluated the bone regeneration potential of RCP particles.

Palatine fissures reside in the central portion of the maxilla in rats [38]. There were three main reasons for using the rat palatine fissure as the graft site in this study. First, the rat palatine fissure is a congenital bone defect in the oral cavity; human alveolar cleft is generated by a congenital malformation with collapse of alveolar segments [5,41]. Second, it anatomically imitates the human alveolar cleft because the palatine fissure is surrounded by bone on two sides—and the remaining two sides are covered with the palatal membrane and the nasal membrane. Third, the potential of bone repair in the maxillary sinus and alveolar ridge could be higher than that in the palatine fissure [42,43], since pre-existing bone exhibits spontaneous healing potential when damaged. Autologous bone chips and β-TCP particles were transplanted into rat palatine fissures in our previous study. In the study, interestingly, there was less bone formation with β-TCP, according to the two dynamic parameters—bone volume and bone mineral density (BMD)—than in autologous bone [38]. This result was in contrast to previous studies, which had generally shown a similar effect regarding maxillary sinus augmentation or edentulous alveolar ridge augmentation using autogenous iliac bone or β-TCP grafts [44,45]. The results suggested that the location of rat palatine fissure could be a feasible model to develop novel bone substitutes, as an alternative to autogenous bone grafts.

In order to find the optimal cross-link density of RCP, three types of RCP particles with similar pore sizes were compared. Pore size is an essential consideration in the development of a bone substitute for bone regeneration. If pores are too small, cell migration is limited. If pores are too large, there is a decrease in surface area, limiting cell adhesion [46]. In vivo results based on micro-CT analysis and histological analysis demonstrated that medium cross-linked RCP particles created a better environment for bone tissue generation compared to low or high cross-linked RCP particles. In addition, the average of bone volume and BMD in medium cross-linked RCP particles were similar to those of autologous bone grafts obtained from our study at 16 weeks. Furthermore, the averages of bone volume and BMD in medium cross-linked RCP particles were higher than those of β-TCP grafts in the study. However, the averages of bone volume and BMD in low or high cross-linked RCP particles were less than those in autologous bone grafts. Medium-cross linked RCP particles are useful for bone reconstruction for alveolar cleft as an alternative to autogenous bone grafts.

It is generally known that modifying the cross-link density of collagen material affects the bioabsorption rate. The bioabsorption rate of a graft material is also indirectly related to its mechanical strength, which ensures that it stays in place at the graft site. Low cross-linked RCP particles led to lower amounts of newly formed bone. This could be explained by the fact that the particles may have biodegraded or been rapidly bioresorbed. In addition, modifying the cross-link density affected the water absorption rate [47,48,49]. In vitro analysis showed that there was a correlation between increased cross-link density and reduced water absorption rate. When a material has a higher water absorption, the contact angle of a hydrophilic material is small. Hydrophilicity of a biomaterial surface is known to have a strong effect on cell response and biocompatibility [50,51,52]. Increased hydrophilicity of the surface of a dental implant has been shown to increase the extent of new bone formation around the dental implant, which is thought to be due to faster cell adhesion and spreading [53,54]. High-cross-linked RCP particles resulted in lower bone formation volume; this may indicate that the water absorption rate may be lower than that of medium-cross-linked RCP particles. However, these assumptions need to be further verified in future studies.

The recruitment of osteoblasts (which are responsible for new bone matrix formation) and osteoclasts (which regulate bone matrix decomposition and calcium reuptake) is known to be an important factor in bone homeostasis [55]. Significant differences in the numbers of ALP+ osteoblasts and TRAP+ osteoclasts were observed between the three types of RCP particles. The medium cross-linked RCP particle was most effective at generating high numbers of osteoblasts and osteoclasts. Hence, medium cross-linked RCP particles may be effective in generating new bone marrow and also maintaining bone homeostasis. The precise mechanism by which medium cross-linked RCP particles enhance bone tissue formation and maintain bone homeostasis was not investigated in this study. Future studies are required to confirm and further evaluate these results.

## 5. Conclusions

This study was the first experiment in which RCP particles were grafted into a rat palatine fissure and bone formation was evaluated. The results demonstrated that medium cross-linked RCP particles were most suitable for bone formation in the palatine fissure. These medium cross-linked particles could potentially be useful for bone reconstruction of the human alveolar cleft. The findings of this study could open new avenues of research into the pathology of the alveolar cleft and facilitate further development of new materials for bone reconstruction.

## Figures and Tables

**Figure 1 materials-14-02306-f001:**
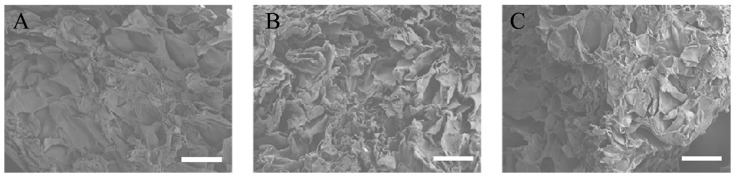
EPMA images of the three types of cross-linked RCP particles. Images taken with a field-emission electron probe microanalyzer (EPMA). Scale bars represent 100 μm. (**A**) EPMA image of low cross-linked RCP particle. (**B**) EPMA image of medium cross-linked RCP particle. (**C**) EPMA image of high cross-linked RCP particle. Three types of RCP particles show similar pore sizes.

**Figure 2 materials-14-02306-f002:**
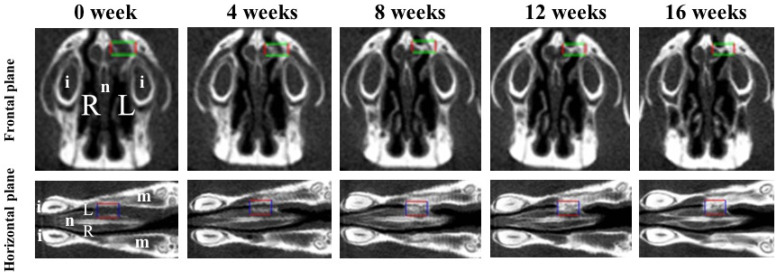
Two viewpoints of micro-CT images of the palatine fissures in the frontal plane and the horizontal plane. Box areas in the frontal plane and the horizontal plane show the palatine fissures in palates of rats. Radiological analysis performed in approximately the same place (box area) at the anatomical location using the regions of interest (ROIs) before grafting (0 week) and 4, 8, 12 and 16 weeks after grafting. The size of ROI is 1.8 × 2.7 × 0.9 mm^3^ (box areas). i: incisor. L: The left palatine fissure (graft side). m: molars. n: nasal septum. R: The right palatine fissure (non-graft side).

**Figure 3 materials-14-02306-f003:**
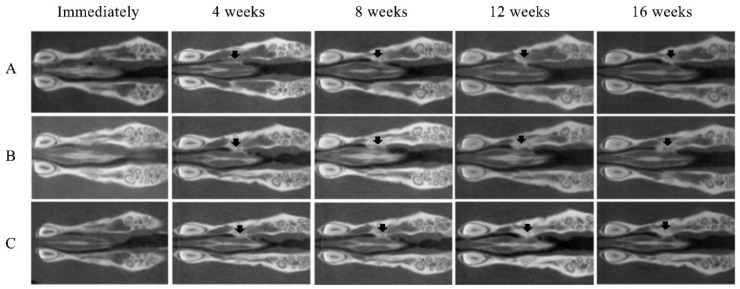
Micro-CT images of the maxilla immediately after grafting and at 4, 8, 12 and 16 weeks after grafting. New bone formation (arrows) at the margins of the left palatine process of the maxilla was noted on the radiographs of all samples. (**A**) Group 1 (low cross-linked RCP particles). (**B**) Group 2 (medium cross-linked RCP particles). (**C**) Group 3 (High cross-linked RCP particles). Each image shows the horizontal plane section of the maxilla in the rats.

**Figure 4 materials-14-02306-f004:**
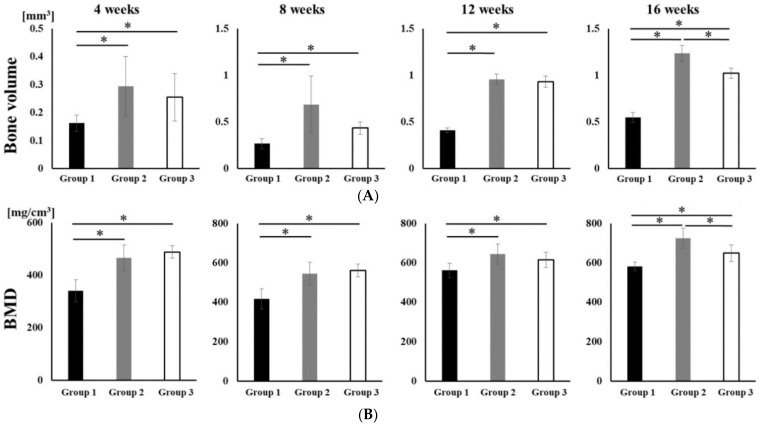
Micro-CT analyses of the palatine fissure at 4, 8, 12 and 16 weeks after grafting. The bone volume and BMD in group 2 were significantly higher than those in group 1 and 3 at 16 weeks after grafting. (**A**) The bone volume of hard tissue in palatine fissures from groups 1, 2 and 3. (**B**) The BMD of the newly formed hard tissue in palatine fissures from groups 1, 2 and 3. * *p* < 0.05. The results are shown as means, with error bars representing the standard deviation (*n* = 15).

**Figure 5 materials-14-02306-f005:**
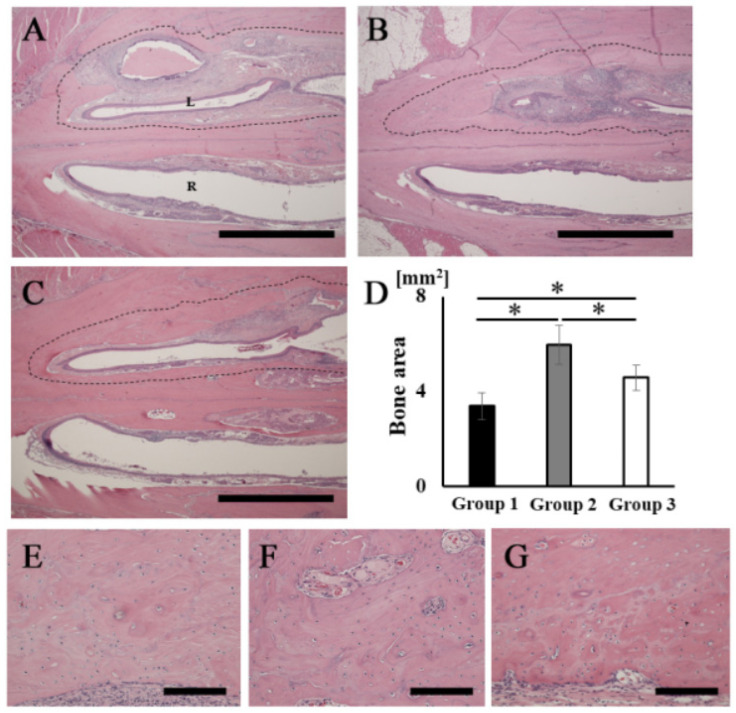
Histological analysis of rat palatine fissures. The horizontal plane section (stained with H&E at 16 weeks after grafting) is shown. (**A**,**E**) Group 1. (**B**,**F**) Group 2. (**C**,**G**) Group 3. (**D**) New bone tissue area (dashed line area) in groups 1, 2 and 3. Newly formed bone areas in bone defects measured using ImageJ software at 16 weeks after grafting. (**A**–**C**) Overview of rat palatine fissure. Scale bars represent 2 mm. (**E**–**G**) Higher magnification of the new bone tissue area in (**A**–**C**), respectively. Scale bars represent 150 μm. L: The left palatine fissure (implantation side). R: The right palatine fissure (normal side). * *p* < 0.05. The results are shown as means, with error bars representing the standard deviation (*n* = 15).

**Figure 6 materials-14-02306-f006:**
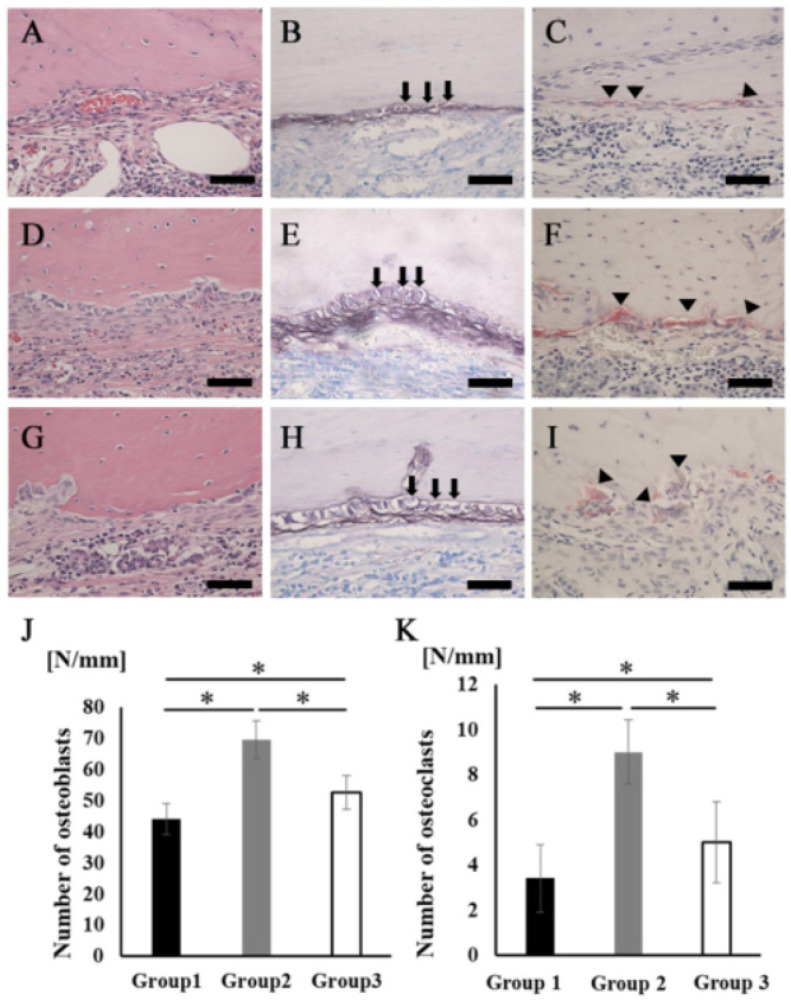
Osteoblasts and osteoclasts in new bone tissue. (**A**–**C**) Group 1. (**D**–**F**) Group 2. (**G**–**I**) Group 3 (**A**,**D**,**G**) Newly formed bone area stained with H&E at higher magnification. Scale bars represent 50 μm. (**B**,**E**,**H**) The images of the area equivalent to (**A**), (**D**), and (**G**) stained with ALP. Scale bars represent 50 μm. The arrows indicate osteoblasts. (**C**,**F**,**I**) The images of the area equivalent to (**A**), (**D**), and (**G**) stained with TRAP. Scale bars represent 50 μm. The arrowheads indicate osteoclasts. (**J**) The number of osteoblasts in the newly formed bone area. (**K**) The number of osteoclasts in the newly formed bone area. * *p* < 0.05. The results are shown as means, with error bars representing the standard deviation (*n* = 15).

**Table 1 materials-14-02306-t001:** Characteristics and pore size of three types of cross-linked RCP particles.

	Low Cross-Linked RCP	Medium Cross-Linked RCP	High Cross-Liked RCP
Cross-linking time (h)	3.5	4.75	7
Water absorption rate (%)	596	575	463
Acid decomposition residual rate (%)	27	42	65
Pore size (µm)	87.2	86.4	86.8

## Data Availability

The data presented in this study are available in insert article.

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
