# Peer review of "A Novel Bone Substitute Based on Recombinant Type I Collagen for Reconstruction of Alveolar Cleft"

_materials, 2021, doi:10.3390/ma14092306_

Round 1

Reviewer 1 Report

The article describes a novel bone substitute containing RGD adhesion motifs in three cross-linked forms and its incorporation in palatine fissure of rat maxilla.

About the article I have some questions and comments:

The first note I have on the form of the article, referring to sentences in the text is very difficult without line numbers.

The article is written clearly and comprehensibly.

Introduction is maybe longer than necessary, but introduces the issue on which the experiment is based.

Paragraph 2.7, the first sentence is meaningless.

Table in Fig1 describes results of measurements and should be in the part 3. Results like a table and not image.

Figure 3 could be interesting, but in such resolution is nothing visible on the fissure part.

Figure 6 is in low quality, if possible, use images with better resolution.

If the previous sections are sufficient, this is not the case in the discussion section. Discussion has to comment the results, but here a big part of discussion is very similar to introduction. The whole first half of the discussion describes the reasons why the material (without cross-linking) and the place of implantation were chosen, which is certainly related to the introduction, but not to the discussion.

The whole topic and the results are very interesting, but they must also be better described in the discussion section and argumented in relation to known results from the literature.

Author Response

Introduction is maybe longer than necessary, but introduces the issue on which the experiment is based.

Response: We agree with reviewer’s proposal.  We revised the manuscript, “Introduction” section according to reviewer’s suggestion.

Paragraph 2.7, the first sentence is meaningless.

Response: Thank you for your valuable suggestion. We agree with your suggestion. The first sentence was removed from the reivsed text.

Table in Fig1 describes results of measurements and should be in the part 3. Results like a table and not image.

Response: Thank you for your valuable suggestion. We agree with your suggestion. We changed Figure 1D to Table 1 in the revised version.

Figure 3 could be interesting, but in such resolution is nothing visible on the fissure part.

Response: Thank you for your valuable suggestion. We added the arrows to show the fissure part in the figure 3 of the revised version.

Figure 6 is in low quality, if possible, use images with better resolution.

Response: Thank you for your valuable suggestion. We revised the images of  ALP staining in the revised version. We did not changes the images of TRAP staining. Osteoclasts can be visibled in the image of TRAP staining.  

If the previous sections are sufficient, this is not the case in the discussion section. Discussion has to comment the results, but here a big part of discussion is very similar to introduction. The whole first half of the discussion describes the reasons why the material (without cross-linking) and the place of implantation were chosen, which is certainly related to the introduction, but not to the discussion.

Thank you for your valuable suggestion. We agree with your suggestion. We revised "Discussion" section according to the suggestion.

Reviewer 2 Report

The presented paper is quite interesting. Ito et al. showed that medium cross-linked recombinant peptide (RCP) particles support bone formation in vivo using a rat model. After 16-weeks of grafting, increase in bone volume and bone mineral in the palatine fissure was observed. Developed RCP particles seem to be a very promising material using in bone tissue engineering. Nevertheless, future studies are required e.g. evaluation of inflammation response, aiming to complete assess medical potential.

Although all I have some questions/remarks to the Authors:

  1. In section 2.2. "Preparation of RCP particles with different densities" lacks reference to the literature. What was the average size of RCP particles?
  2. What was the aim of conducting acid-decomposition residual rate experiment? Authors did not mention it in the manuscript.
  3. Were RCP particles pre-soaked in blood plasma or in saline solution before implantation into the rat palatine fissure?
  4. Water absorption rate of three types of RCP particles was in the range of 463-595 %. At what time was it achieved? Did you not observed swelling of RCP particles? Did you weigh RCP particles before and after the water absorption rate experiment?
  5. Authors wrote: “EPMA images confirmed the wide range of pore sizes in the three types of RCP particles and revealed that many of the smaller pores were not interconnected (Figure 1A–C)”. To assess pores distribution and their interconnectivity within the structure of the material, cross-section images obtained by micro-computed tomography is needed. EPMA images do not clearly show that.
  6. Figure 6 shows the number of osteoblasts and osteoclasts in the newly formed bone area. Unit N/mm means a number of cells per mm2? What program did you use to count cells?
  7. In Section 3.1 Authors wrote: “No wound dehiscence, severe inflammation, or swelling was observed in any of the samples throughout the experimental period”. How did you evaluate inflammation response during in vivo study? Did you assess the level of pro-inflammatory cytokines/factors in the rat blood?

Author Response

  1. In section 2.2. "Preparation of RCP particles with different densities" lacks reference to the literature. What was the average size of RCP particles?  Response: Thank you for your valuable question. We described the average size of RCP in the Materials and method sections of the revised version. 
  2. What was the aim of conducting acid-decomposition residual rate experiment? Authors did not mention it in the manuscript. Response: Thank you for your valuable suggestion. We agreed with your suggestion.                                                                                       The reason why we mentioned the acid-decomoposition residual rate is to provide the data as the characteristics of each RCP particle. Actually, we do not think that the acid-decomposition residual rate was not influeced to the hard tissue formation. Therefore, we only mentioned their results. 
  3. Were RCP particles pre-soaked in blood plasma or in saline solution before implantation into the rat palatine fissure? Response: Thank you for your valuable question. We did not pre-soaked RCP particles in any liquid. We implanted the RCP particles under dry condition. 
  4. Water absorption rate of three types of RCP particles was in the range of 463-595 %. At what time was it achieved? Did you not observed swelling of RCP particles? Response: Thank you for your valuable question. We tested that the substance was saturated with absorbled water. So we mentioned that in the Materials and method section in the revised version. Did you weigh RCP particles before and after the water absorption rate experiment? 
  5. Response: Thank you for your valuable question. We did not weighe before and after the experiment because we implanted the RCP particles under dry condition. 
  6. Authors wrote: “EPMA images confirmed the wide range of pore sizes in the three types of RCP particles and revealed that many of the smaller pores were not interconnected (Figure 1A–C)”. To assess pores distribution and their interconnectivity within the structure of the material, cross-section images obtained by micro-computed tomography is needed. EPMA images do not clearly show that. Response: Thank you for your valuable question. We agree with your suggestion. We think the cross section image when the RCP particle have the interconnectivity. However, in fact, the RCP particles do not have interconnectivity within the particles. Therefore, we do not think that the cross-section images do not need to our manuscript.
  7. Figure 6 shows the number of osteoblasts and osteoclasts in the newly formed bone area. Unit N/mm means a number of cells per mm2? What program did you use to count cells? Thank you for your valuable question. We did not use the program to count cells. The number of the cells was counted by the naked eye from a few researchers.
  8. In Section 3.1 Authors wrote: “No wound dehiscence, severe inflammation, or swelling was observed in any of the samples throughout the experimental period”. How did you evaluate inflammation response during in vivo study? Did you assess the level of pro-inflammatory cytokines/factors in the rat blood? Thank you for your valuable question. We did not any test to evaluate the inflammation response. Oral surgens in our team visually inspected the reddening, swelling,etc. for inflammation response.

Reviewer 3 Report

The manuscript entitled:” A novel bone substitute based on recombinant typeI collagen for reconstruction of alveolar cleft” describes the research results they have found with  3 different types of cross-linked RCP particles. Some points are not clear for me and needs to be modified.

    1.Why are rats of 11 weeks old used?  After the test period they are 7          months old and adult, whereas in human most of the alveolar cleft reconstructions take place in the first years of life, during childhood. Is it not better to start with younger animals?

  1. The figure legends are very short and not telling what is visible in the figures. And very difficult to follow.
  2. Figure 3 shows microCT pictures but in my opinion they are not all in the same plane.For instance in figure B of timepoint 0 it seems that much more bone is present than in B after 4 weeks. This is also due for some other timepoint and types of RCP.
  3. In figure 5 the scale bar in figure A,B,C is 150micrometer=0.15 mm. In figure E,F,G the scale bar is 2 mm. This means that the magnification in A,B,C is higher. But the legend says otherway around and the pictures shows too. Most of the time the size of the scalebar is consistant and only micrometer is different and not both different.
  4. In figure 6 osteoblasts and osteoclasts were stained resp. with ALP and TRAP. The staining is not very clear and it is not clearly visible that in group 2 the osteoblasts are cuboidal and in the others not. Also in osteoclast pictures osteoclasts are not visisble and nuclei not stained. Therefor I cannot see that multinucleated osteoclasts are present. The text also says that more osteoblasts and osteoclasts are present in group 2, but this is not visible in these figures. All figures have 3 arrows/arrowheads.

Author Response

  1.   1.Why are rats of 11 weeks old used?  After the test period they are 7 months old and adult, whereas in human most of the alveolar cleft reconstructions take place in the first years of life, during childhood. Is it not better to start with younger animals?    Response: Thank you for your valuable comments. We agree with your comments. On the other hand, in this study, our aim is to comapare the potential for bone tissue formation in three types of RCP particles. Therefore, we choosed 11 weeks rats because it is easy to approach to palatin fissure in the rats with 11 weeks old. In addition, it is easy to evaluate the bone volume and BMD in the palatine fissure because the space of palatine fissure is reasonable.  
  2. The figure legends are very short and not telling what is visible in the figures. And very difficult to follow. 

    Response: Thank you for your valuable comments. We agree with your comments. We revised the figure legends in the re-revised text.

  3. Figure 3 shows microCT pictures but in my opinion they are not all in the same plane.For instance in figure B of timepoint 0 it seems that much more bone is present than in B after 4 weeks. This is also due for some other timepoint and types of RCP.  Thank you for your valuable comments. We agree with your comments. In fact, we observed and obtained their images from the same plane. However, due to the growth of the rats, the weight of the rats gains after transplantation. Therefore, there is a slight discrepancey betweeen before and after transplantation. Therefore, we revised the sentence in the revised verstion.
  4. In figure 5 the scale bar in figure A,B,C is 150micrometer=0.15 mm. In figure E,F,G the scale bar is 2 mm. This means that the magnification in A,B,C is higher. But the legend says otherway around and the pictures shows too. Most of the time the size of the scalebar is consistant and only micrometer is different and not both different. Response: Thank you for your comments and we are sorry for that the description was mistaken. We revised the descriotion of their scale bars.
  5. In figure 6 osteoblasts and osteoclasts were stained resp. with ALP and TRAP. The staining is not very clear and it is not clearly visible that in group 2 the osteoblasts are cuboidal and in the others not. Also in osteoclast pictures osteoclasts are not visisble and nuclei not stained. Therefor I cannot see that multinucleated osteoclasts are present. The text also says that more osteoblasts and osteoclasts are present in group 2, but this is not visible in these figures. All figures have 3 arrows/arrowheads.Response: Thank you for your valuable suggestion. We agree with your suggestion. We tried it again and obtained the high resolution images. Therefore, we replaced the images for ALP staining.

Round 2

Reviewer 1 Report

Some recommended changes have been made. Some others, in the response described as "has been made according to suggestion" are in the revised manuscript in the same form as before the revision.

E.g.

>Table in Fig1 describes results of measurements and should be in the part 3. Results like a table and not image.

Response: Thank you for your valuable suggestion. We agree with your suggestion. We changed Figure 1D to Table 1 in the revised version.

  • Table is still Figure 1D

>Figure 3 could be interesting, but in such resolution is nothing visible on the fissure part.

Response: Thank you for your valuable suggestion. We added the arrows to show the fissure part in the figure 3 of the revised version.

  • The Figure 3 seems to be the same and no arrows has been added

The discussion section has been partially improved by adding a few new paragraphs.

Author Response

Table in Fig1 describes results of measurements and should be in the part 3. Results like a table and not image.

Response: Thank you for your valuable suggestion. We agree with your suggestion. We changed Figure 1D to Table 1 in the revised version.

  • Table is still Figure 1D

Response: Thank you for your indication. We are sorry for that. We are sure we changed Figure 1D to Table 1 in the revised template.

  • The Figure 3 seems to be the same and no arrows has been added.

Response: Thank you for your indication. We are sorry for our mistakes.  We are sure we added the arrows to show the fissure part in the figure 3 in the template version. 

  • The discussion section has been partially improved by adding a few new paragraphs.

Response: Thank you for your understanding.  

Reviewer 3 Report

The legends at the figures are much more clear now, but now in Figure 2 more in the legend than in the figure. I do not see any box areas in the frontal plane and also the i:incisor; m:molars and n: nasal septum is not visible in Figure 2 for me. I think when these letters are added the manuscript is ready to be accepted.

Author Response

The legends at the figures are much more clear now, but now in Figure 2 more in the legend than in the figure. I do not see any box areas in the frontal plane and also the i:incisor; m:molars and n: nasal septum is not visible in Figure 2 for me. I think when these letters are added the manuscript is ready to be accepted.

Response: Thank you  for your understanding and indication. We added abox areas, i, m, and n in Figure 2 in the template version according to your indication.